# miR-135a Suppresses Granulosa Cell Growth by Targeting *Tgfbr1* and *Ccnd2* during Folliculogenesis in Mice

**DOI:** 10.3390/cells10082104

**Published:** 2021-08-17

**Authors:** Lei Wang, Yaru Chen, Shang Wu, Jinhua Tang, Gaogui Chen, Fenge Li

**Affiliations:** 1Key Laboratory of Swine Genetics and Breeding of Ministry of Agriculture and Rural Affairs & Key Laboratory of Agricultural Animal Genetics, Breeding and Reproduction of Ministry of Education, Huazhong Agricultural University, Wuhan 430070, China; wanglei1990@webmail.hzau.edu.cn (L.W.); chenyaru@webmail.hzau.edu.cn (Y.C.); wushang@webmail.hzau.edu.cn (S.W.); inhuaTang@webmail.hzau.edu.cn (J.T.); CGG@webmail.hzau.edu.cn (G.C.); 2The Cooperative Innovation Center for Sustainable Pig Production, Wuhan 430070, China

**Keywords:** mice, miR-135a, cell growth, *Tgbf1*, *Ccnd2*, ESR2, folliculogenesis

## Abstract

The success of female reproduction relies on high quality oocytes, which is determined by well-organized cooperation between granulosa cells (GCs) and oocytes during folliculogenesis. GC growth plays a crucial role in maintaining follicle development. Herein, miR-135a was identified as a differentially expressed microRNA in pre-ovulatory ovarian follicles between Large White and Chinese Taihu sows detected by Solexa deep sequencing. We found that miR-135a could significantly facilitate the accumulation of cells arrested at the G1/S phase boundary and increase apoptosis. Mechanically, miR-135a suppressed transforming growth factor, beta receptor I (*Tgfbr1*) and cyclin D2 (*Ccnd2*) expression by targeting their 3′UTR in GCs. Furthermore, subcellular localization analysis and a chromatin immunoprecipitation-quantitative real-time PCR (ChIP-qPCR) assay demonstrated that the TGFBR1-SMAD3 pathway could enhance *Ccnd2* promoter activity and thus upregulate *Ccnd2* expression. Finally, estrogen receptor 2 (ESR2) functioned as a transcription factor by directly binding to the miR-135a promoter region and decreasing the transcriptional activity of miR-135a. Taken together, our study reveals a pro-survival mechanism of ESR2/miR-135a/*Tgfbr1*/*Ccnd2* axis for GC growth, and also provides a novel target for the improvement of female fertility.

## 1. Introduction

Folliculogenesis, from primordial follicle activation to oocyte release, is a complex and dynamic process that relies on synchronization between the oocyte maturation and the neighboring granulosa cell growth. During this process, ovarian granulosa cells (GCs) provide necessary nutrients and steroids to the oocytes, and therefore play vital roles in ovarian follicle development [1]. In the awakening stage of mammalian oocytes, the flattened GCs from primordial follicles first proliferate and differentiate into cuboidal granulosa cells. This is followed by the dramatic growth of oocytes [2]. Subsequently, GCs rapidly proliferate to form an antral structure composed of multilayer granulosa cells and differentiate into mural GCs and cumulus GCs [3,4]. Only a subset of the antral follicles responds to LH and further releases fertilizable oocytes. The remaining GCs in the raptured follicle will undergo terminal differentiation and form the corpus luteum, which secretes progesterone necessary to maintain pregnancy [5]. GC growth is necessary for follicular maturation and ovulation, whereas GC apoptosis would lead to follicular atresia and degradation [6,7,8].

MicroRNAs (miRNAs) are small, noncoding RNAs that negatively regulate gene expression post-transcriptionally [9]. Currently, miRNAs have been suggested to regulate GC functions and thus closely participate in folliculogenesis, oogenesis, and steroidogenesis [10]. MiR-143 plays a critical role in follicular atresia by regulating cell apoptosis and steroidogenesis [11], and miR-126-3p promotes cell proliferation in the porcine granulosa cells [12]. MiR-16 promotes ovarian GC proliferation and inhibits GC apoptosis in polycystic ovarian syndrome [13]. Notably, miR-135a (as a tumor suppressor) is involved in regulating cell cycle and cell proliferation [14]. The upregulation of miR-135a promotes cell apoptosis and inflammation, along with inhibited cell proliferation and decreased macrophage autophagy [15]. Furthermore, miR-135a participates in the regulation of cyclin E1 (*CCNE1*) expression and the accumulation of cells arrested at the G1/S phase boundary [16]. In addition, hsa-miR-135 is associated with the survival of patients who have serous ovarian carcinoma, and thus may serve as a potential prognostic biomarker [17]. Our previous work identified that the expression of miR-135a was significantly different in pre-ovulatory ovarian follicles of Large White (LW) and Taihu sows detected by Solexa deep sequencing [18]. Moreover, miR-135a promotes apoptosis and the DNA damage response in GCs in polycystic ovary syndrome [19]. Therefore, investigating the role of miR-135a in female reproduction is indispensable

Transforming growth factor (TGF)-β signaling activates its own membrane serine/threonine kinase receptors TGFBR2 (type II receptor) and TGFBR1 (type I receptor) to promote the binding of SMAD family member (SMAD) 2/3 intracellular signaling to SMAD4 in the nucleus. Then, the SMAD complex regulates transcription by binding to the targeted promoter regions referred to as SMAD-binding elements (SBEs) [20]. TGF-β signaling pathway is crucial for ovarian granulosa cell growth and female fertility [21]. FSH and LH could regulate natriuretic peptide C (NPPC) levels via regulating the levels of TGFB1, TGFBR2, and TGF-β downstream SMAD proteins in GCs and controlling the process of oocyte meiosis [22]. On the other hand, miRNAs can regulate GC function and follicle development by targeting TGF-β signaling pathway. MiR-424/503 cluster members modulate bovine granulosa cell proliferation and cell cycle by targeting SMAD7 [23]. MiR-181a regulates porcine GC apoptosis by targeting TGFBR1 [24], and miR-130a/TGF-β1 axis is involved in sow fertility by regulating granulosa cell apoptosis [25].

In our study, we determined the role of miR-135a in mGC growth retardation and identified *Tgfbr1* and *Ccnd2* as the targets of miR-135a. In addition, estrogen-induced upregulation of *Tgfbr1* and *Ccnd2* mRNA levels attributed to the binding of ESR2 to canonical estrogen response element (ERE) in miR-135a promoter. In conclusion, our study provides new insight into understanding the network of ESR2/miR-135a/TGFBR1/CCND2 in mGC growth and follicle development.

## 2. Materials and Methods

### 2.1. Animals and Collection of Ovaries

All mice were housed in a pathogen-free environment at 20–22 °C, 50–70% relative humidity, and under a 12 h/12 h light/dark cycle. All mice had *ad libitum* accesses to a standard chow diet. Twenty-one-day-old female Kunming mice (Center for Disease Control; Hubei, China) were injected intraperitoneally (i.p.) with 10 units of pregnant mare serum gonadotropin (PMSG; Ningbo Second Hormone Factory, Ningbo, China). After 48 h, mice were killed by cervical dislocation, and their ovaries were harvested for in vitro experiments. All animal experiments were conducted in accordance with the guidelines of the Animal Care and Ethics Committee of Huazhong Agricultural University.

### 2.2. Plasmid Construction

Full-length *Tgfbr1* (NM_001312868.1) was cloned into *pcDNA3.1* to generate *pcDNA3.1-**Tgfbr1* and full-length *Ccnd2* (NM_009829.3) was cloned into *pcDNA3.1* to generate *pcDNA3.1-Ccn**d2* with Trelief™ SoSoo Cloning Kit (TSV-S2, Tsingke Biotechnology, Wuhan, China). Site-directed mutagenesis for the ESR2 binding site was performed using the primers which were synthesized by Tsingke Biotechnology and described in Appendix A. Truncated fragments of the mouse miR-135a promoter were amplified from mouse genomic DNAs and cloned into the pGL3-basic vector.

### 2.3. Cell Culture and Cell Transfection

Primary murine granulosa cells (mGCs) were isolated from ovarian follicles and cultured as described previously [26]. Murine granulosa cells were cultured in Dulbecco’s minimum essential medium/nutrient (DMEM)/F-12 (11320033, Gibco, CA, USA) supplemented with 10% fetal bovine serum (10099141C, Gibco, CA, USA), 100 unit/mL penicillin and 100 mg/mL streptomycin (15140122, Gibco, CA, USA) at 37 °C in a humidified atmosphere of 5% CO_2_. Cells were seeded in plates and grew up to 70–90% confluent at the time of transfection. The plasmids, miRNAs and siRNAs were transfected using Lipofectamine™ 3000 (L3000015, Invitrogen™, Carlsbad, CA, USA) and RNAiMAX transfection reagent (13778030, Invitrogen™, CA, USA), respectively, as described previously [18,27].

### 2.4. Luciferase Reporter Assay

Each recombinant construct plasmids were transfected with *pRL-TK* (E2241, Promega, Wisconsin, USA) into mGCs. After 24 h, cells were collected and luciferase activities were measured using the Dual-Luciferase Reporter Assay System (E1910, Promega, Wisconsin, USA) according to the manufacturer’s instructions. The experiments were repeated at least three times, and the results were expressed as the means ± SD.

### 2.5. Quantitative Real-Time PCR

Total RNA and cDNA were prepared from cultured mGCs as described previously [18]. Quantitative real-time PCR was performed using the iTaq^TM^ Universal SYBR Green Super Mix (172-5121, Bio-Rad, Hercules, CA, USA) and analyzed on CFX384 Touch™ Real-Time PCR Detection System (Bio-Rad, Hercules, CA, USA). The primers are listed in Appendix A.

### 2.6. Western Blot

Total protein was lysed by RIPA Lysis Buffer (P0013B, Beyotime, Shanghai, China) with 1% PMSF (ST505, Beyotime, Shanghai, China) and 1% Phosphatase Inhibitor Cocktail I (HY-K0021, MedChemExpress, Shanghai, China). The whole extracts were analyzed using sodium dodecyl sulfate polyacrylamide gel electrophoresis (SDS-PAGE) (G2003-50T, Servicebio, Wuhan, China) and transferred to polyvinylidene difluoride membrane (HATF09025, Millipore, Massachusetts, USA) by electroblotting. Then, the membranes were blocked with 5% non-fat dried milk or 3% BSA in TBST (20 mmol/L Tris-HCl, pH 7.5, 150 mmol/L NaCl, 0.1% Tween-20) and then incubated with primary antibodies specific for CCND2 (1:1000, A1773, Abclonal, Wuhan, China), TGFBR1 (1:1000, A16983, Abclonal, Wuhan, China), SMAD3 (1:1000, A19115, Abclonal, Wuhan, China), p-SMAD3 (1:1000, C25A9, Cell signaling, Danvers, MA, USA), β-actin (1:10000, AC026, Abclonal, Wuhan, China), CDK4 (1:1000, A0366, Abclonal, Wuhan, China), CCNE1 (1:1000, A14225, Abclonal, Wuhan, China), BAX (1:2000, A19684, Abclonal, Wuhan, China), BCL2 (1:1000, A19693, Abclonal, Wuhan, China), PCNA (1:2000, A12427, Abclonal, Wuhan, China), and HA-tag (1:2000, ab9110, Abcam, Cambridge, UK) overnight at 4 °C. The membranes were incubated at room temperature for 2 h with a diluted (1:3000) secondary antibody against rabbit (1:5000, AS014, Abclonal, Wuhan, China) or mouse (1:5000, AS003, Abclonal, Wuhan, China) primary antibodies. An Image Quant LAS4000 mini (GE Healthcare Life Sciences, Piscataway, NJ, USA) was used to detect protein expression.

### 2.7. Chromatin Immunoprecipitation

pCMV-N-HA-ESR2 was constructed and transfected into mGCs for a Chromatin immunoprecipitation (ChIP) assay. ChIP was performed using the EZ-ChIP Kit (Millipore, MA, USA) according to the manufacturer’s instructions. The AVCX130 system (Sonics & Materials, Newtown, CT, USA) was used for cell sonication. Anti-HA (ab9110, Abcam), anti-SMAD3 (ab208182, Abcam) and normal anti-mouse-IgG (ab6789, Abcam, Cambridge, UK) were used for the immunoprecipitation reactions. DNA fragment from ESR2-immunoprecipitated complex was quantified via qPCR. The primer sequences are described in Appendix A.

### 2.8. Flow Cytometry Analyses

Fluorescence-activated cell sorting (FACS) was used to measure cell cycle and apoptosis. For the analysis of cell cycle, cells stained with Propidium Iodide (PI) according to the manufacturer’s manual. The apoptosis experiments were performed according to the manufacturer’s protocol for the Annexin V-FITC Apoptosis Detection Kit (AD10, Dojindo, Shanghai, China).

### 2.9. Cell Viability Assay

Cell viability was determined using the Cell Counting Kit-8 (CCK-8; RM02823, Abclonal, Wuhan, China) assay according to the manufacturer’s instructions.

### 2.10. Immunofluorescence

Our experimental procedures were performed as described previously [26]. Cells were sequentially incubated with primary antibody for anti-Ki67 (1:150, A2076, Abclonal, Wuhan, China) followed by incubation with CY3-conjugated secondary antibodies (1:200, AS007, Abclonal, Wuhan, China) and then stained with DAPI. Finally, images were taken under a microscope (OLYMPUS, Tokyo, Japan).

### 2.11. In Silico Sequence Analysis

The potential targets of miR-135a were predicted using Targetscan (http://www.targetscan.org (accessed on 15 June 2020)). The predicted transcription factors for *Ccnd2* and miR-135a were performed using BIOBASE (http://gene-regulation.com (accessed on 1 August 2020)).

### 2.12. Statistical Analysis

All results are presented as the mean ± SD. Each treatment had three replicates. Two-tailed t-test was used when two groups were compared. Significant differences were evaluated using an independent-samples t-test. *p* < 0.05 was considered statistically significant.

## 3. Results

### 3.1. MiR-135a Inhibits Cell Cycle and Proliferation in Murine GCs

Our previous study has identified that the lower level of miR-135a in pre-ovulatory ovarian follicles of Taihu sows with characteristics of high ovulation rate and large litter size, compared with that of LW sows by Solexa deep sequencing (Appendix A) [18]. To investigate the role of miR-135a in mGC growth, miR-135a overexpression or inhibition was performed in mGCs (Appendix A). Ki67 staining and CCK-8 assay demonstrated that miR-135a overexpression in mGCs resulted in a reduction of Ki67-positive cells and a decline of cell vitality, which was in accordance with the observation of high proliferative activity and cell vitality in miR-135a inhibited mGCs (Figure 1A–C). Moreover, fluorescence-activated cell sorting (FACS) analysis revealed that miR-135a overexpression obviously increased the cell population of the G1 phase, accompanied by an elevated rate of G1/S and a decreased proliferation index (PI, PI = (G2 + S)/G1) in mGCs (Figure 1D). However, miR-135a inhibition reduced the number of GCs in the G1 phase and improved the number of mGCs in the S phase, accompanied by decreased G1/S arrest and increased PI in mGCs (Figure 1D). Subsequently, western blot analysis showed that miR-135a overexpression significantly repressed the protein levels of cyclin-dependent kinase 4 (CDK4), cyclin E1 (CCNE1) and proliferating cell nuclear antigen (PCNA) that were the important regulators of cell cycle and cell proliferation (Figure 1E), while miR-135a inhibition could upregulate these protein levels (Figure 1F). These results suggest that miR-135a may be involved in cell cycle arrest at the G1/S phase. On the other hand, cell apoptosis assay demonstrated a higher apoptosis rate in miR-135 overexpressed mGCs and a lower apoptosis rate in miR-135 inhibited mGCs (Figure 1G). Consistently, miR-135a overexpression increased BAX protein level and decreased BCL2 protein level, which was inverse in mGCs with miR-135a inhibition (Figure 1H,I). These results suggest that miR-135a can repress cell proliferation and accelerate apoptosis in mGCs.

### 3.2. MiR-135a Directly Targets Tgfbr1 and Ccnd2 Gene in mGCs

Targetscan was used to detect potential targeted genes of miR-135a. As a result, *Tgfbr1* and *Ccnd2* were predicted to be the targets of miR-135a (Figure 2A,B). In addition, the miR-135a-binding seed sequences in the *Tgfbr1* 3′-UTR and *Ccnd2* 3′-UTR were also highly conserved in mammals, respectively (Figure 2A,B). Then, we constructed the dual-luciferase reporter vectors containing the wild-type and mutant miR-135a-binding site in the 3′UTR of *Tgfbr1* and *Ccnd2* to analyze the interaction between miR-135a and *Tgfbr1* or *Ccnd2* (Figure 2C,D). Luciferase assay showed that the decreased luciferase activity was observed in mGCs co-transfected with miR-135a mimics and pmirGLO-*Tgfbr1*-3′-UTR-WT or pmirGLO-*Ccnd2*-3′-UTR-WT (Figure 2E,F). However, luciferase activity remained unchanged in mGCs co-transfected with miR-135a mimics and pmirGLO-*Tgfbr1*-3′-UTR-MUT or pmirGLO-*Ccnd2*-3′-UTR-MUT (Figure 2E,F). Then, western blot and qPCR revealed that miR-135a significantly decreased *Tgfbr1* and *Ccnd2* expression (Figure 2G-L). These data suggest that miR-135a directly targets *Tgfbr1* and *Ccnd2* genes and suppresses their expression.

### 3.3. MiR-135a Modulates mGC Proliferation by Targeting Tgfbr1

To investigate the effect of TGFBR1 on mGC growth, the *Tgfbr1* overexpressed vector was transfected into mGCs. Ki-67 staining and CCK-8 assay demonstrated that TGFBR1 could increase cell viability and the population of Ki-67 positive mGCs (Figure 3A,B). Then, FACS analysis demonstrated that *Tgfbr1* overexpression elevated the cell number in S and G2 phases accompanied by decreased G1/S arrest and increased PI in mGCs (Figure 3C). Consistently, CDK4, CCNE1, and PCNA protein levels were increased in *Tgfbr1*-overexpressed mGCs compared with control mGCs (Figure 3D). Moreover, inhibition of endogenous miR-135a expression could increase cell viability, but this effect disappeared after *Tgfbr1* knockdown in mGCs (Figure 3E). In addition, we found that CCND2 enhanced cell viability (Figure 3F) and *Ccnd2* knockdown abolished the function of miR-135a inhibitor on cell viability (Figure 3G). These results suggest that miR-135a can repress mGC proliferation via *Tgfbr1* and *Ccnd2*. In addition, *Tgfbr1* overexpression resulted in a decrease of apoptotic cell rate with 1.5-fold reduction of BAX protein level and 1.3-fold elevation of BCL2 protein level, compared with the control group (Figure 3H,I).

### 3.4. TGFBR1-SMAD3 Signaling Pathway Promotes mGC Proliferation by Regulating Ccnd2 Expression

To investigate whether TGFBR1 is involved in cell cycle and cell proliferation via TGFβ signaling pathway, western blot analysis was used to detect TGFBR1-mediated phosphorylation of SMADs (p-SMADs). *Tgfbr1* overexpression obviously increased p-SMAD3 protein level in mGCs (Figure 4A), whereas miR-135a expression could decrease p-SMAD3 protein level in mGCs (Figure 4B). It has been suggested that p-SMAD3 can translocate into the nucleus from the cytoplasm and then activate the transcriptional expression of downstream genes [28]. Subsequently, immunofluorescence assay showed *Tgfbr1* overexpression promoted the nuclear distribution of SMAD3, while miR-135a overexpression mainly resulted in nuclear exclusion of SMAD3 (Figure 4C). Furthermore, a SMAD3 binding site (−794 bp to −782 bp) was predicted in the *Ccnd2* promoter region by BIOBASE software. ChIP-qPCR analysis revealed that *Tgfbr1* overexpression significantly increased the binding of SMAD3 to the promoter of *Ccnd2* (Figure 4D). Indeed, the mRNA and protein levels of *Ccnd2* were increased in *Tgfbr1*-overexpressed mGCs (Figure 4E,F). These results suggest that TGFBR1 regulates the transcription of *Ccnd2* by facilitating the recruitment of SMAD3 to *Ccnd2* promoter.

### 3.5. Transcription Factor ESR2 Negatively Regulates miR-135a Expression

To identify the regulatory elements of miR-135a expression in mGCs, a series of truncated miR-135a promoters were used to drive luciferase gene expression to detect promoter activity. Notably, the luciferase assay showed that pGL3-miR-135a-D5 had the highest luciferase activity, suggesting that the fragment from -250 bp to -100 bp in the 5′ flanking region contained negative regulatory elements for miR-135a promoter activity (Figure 5A). Subsequently, two ESR2 binding sites (−197 bp~−183 bp, −141 bp~−127 bp) were predicted in the miR-135a promoter region by the BIOBASE software (Figure 5B). To identify ESR2 binding sites, the mutant pGL3-miR-135a-D5 vectors were constructed by site-directed mutagenesis. As a result, compared with the wild-type pGL3-miR-135a-D5, two single ESR2-element mutants and the double ESR2-element mutant showed obvious increases in promoter activity, which suggests that ESR2 could bind to the miR-135a promoter at two sites (−197 bp~−183 bp, −141 bp~−127 bp) and repress miR-135a promoter activity in mGCs (Figure 5C). ChIP-qPCR analysis also demonstrated that ESR2 could specifically bind to the miR-135a promoter region (Figure 5D).

Then, the luciferase assay showed that ESR2 could significantly decrease pGL3-miR-135a-D4-driven luciferase activity, and further downregulate endogenous miR-135a expression in mGCs (Figure 5E,F). As ESR2 can regulate gene expression via canonical estrogen-dependent transcriptional activation, we investigated the effect of estrogen on miR-135a, *Tgfbr1* and *Ccnd2* expression. Upon estrogen stimulation, miR-135a level was significantly decreased by 3-fold, and *Tgfbr1* and *Ccnd2* mRNA levels were increased by 1.6- and 2.1-fold, respectively (Figure 5G). These results suggest that estrogen promotes the binding of ESR2 to the miR-135a promoter region and thus negatively regulates its endogenous expression to improve *Tgfbr1* and *Ccnd2* expression.

## 4. Discussion

The success of female reproduction relies on high quality oocytes which is determined by the well-organized process of follicle development. In the whole process, granulosa cells are mainly responsible for protecting and supporting the oocyte through providing essential nutrients, growth factors, and steroids [29,30]. Therefore, normal GC growth plays a crucial role in maintaining follicle development. In our study, we revealed that miR-135a was a proliferation repressor, resulting in cell cycle arrest at G1 phase in mGCs. Mechanically, we confirmed that miR-135a could target the 3′UTR of *Tgfbr1* and *Ccnd2*. Moreover, TGFBR1-SMAD3 pathway enhanced endogenous *Ccnd2* expression by promoting the transcriptional activity of *Ccnd2*. Furthermore, ESR2 could negatively regulate miR-135a expression via binding ERE element in miR-135a promotor region in mGCs (Figure 6).

It has been suggested that miR-135a is closely related to the negative regulation of TGFβ-SMAD signaling pathway. miR-135a suppresses TGF-β-mediated epithelial-mesenchymal transition by targeting *SMAD3* [31] and *SMAD5* [32]. Moreover, miR-135a-5p is a key regulator of the TGFBR1/TAK1 pathway, resulting in the attenuation of vascular inflammation in rats with chronic kidney disease [33]. Our study identified that miR-135a could suppress the expression of *Tgfbr1* via targeting its 3′UTR. Previous studies reported that TGFBR1 was regulated by several miRNAs [24,34,35]. Both miR-1343 and miR-181a could increase GC apoptosis by targeting *Tgfbr1* [11,34]. miR-140-5p downregulates *Tgfbr1* expression, causing cell cycle arrest at the G1/S phase [35]. Consistently, our study found that miR-135a could result in G1/S arrest and cell proliferation suppression in mGCs, and miR-135a could regulate TGFBR1-mediated cell proliferation. These results provide plenty of evidence to verify that miR-135a represses cell cycle and cell proliferation via targeting *Tgfbr1* in mGCs.

CCND2 is a specific cell cycle regulator during G1/S transition [36] and a proliferation regulator during folliculogenesis [37]. The CCND2-CDK4 complex phosphorylates retinoblastoma-associated protein and regulates the cell cycle during G1/S transition. Phosphorylation of RB1 allows dissociation of the transcription factor E2F from the RB/E2F complex and subsequently activates transcription of E2F-targeted genes which are responsible for cell cycle [38]. CCND2 expression is also regulated by many microRNAs. MiR-206 inhibits cell proliferation by targeting *Ccnd2* and retarding G1/S phase transition in human laryngeal squamous cells [39]. MiR-373-3p inhibits cell propagation and boosts apoptosis in gemcitabine resistance pancreatic carcinoma cells by targeting *Ccnd2* [40]. In this study, we found that miR-135a could repress *Ccnd2* expression by targeting its 3′UTR and regulate CCND2-mediated mGC proliferation. As CCND2 is widely known as a cyclin, our study did not further explore its roles in cell cycle and proliferation.

Binding of TGF-β to its receptor leads to activation of the transcription factor SMAD3, and SMAD3 translocates into the nucleus where the factors can induce transcriptional expression of targeted genes [41,42,43]. Our results showed *Tgfbr1* overexpression significantly increased but miR-135a overexpression obviously decreased the expression level of p-SMAD3 and nuclear distribution of SMAD3. It has been reported that SMAD3 co-precipitates with upstream DNA sequences of *Ccnd2* transcription start sites in mGCs [44]. Indeed, our study found *Tgfbr1* overexpression could enhance endogenous *Ccnd2* expression via binding of SMAD3 to the *Ccnd2* promoter, whereas the function of TGFBR1 was further suppressed in miR-135a overexpressed mGCs. These data suggest that miR-135a inhibits *Ccnd2* expression via the targeted mRNA inhibition and suppression of TGF-β signaling.

miRNA expression is also regulated by transcription factors in granulosa cells. Steroidogenic factor-1 suppresses miR-383 transcription and then mediates estradiol release in GCs [45]. TGF-β1 enhances the binding of p53 and NF-kB to the promoter region of the miR-244 host gene, promotes pri-miR-244 transcription, and affects GC proliferation and estradiol synthesis in GCs [46]. ESR2 can regulate its targeted genes through the canonical estrogen response element (ERE)-dependent transcriptional activation or through regulatory pathways independent of binding to the cognate response element [47]. Loss of ESR2 in mutant mice, resulting in diminished GC-responsiveness to gonadotropins, reduced estrogen production associated with impaired follicle maturation and ovulation failure [48]. Our study identified that ESR2 negatively regulated miR-135a expression through binding to two ERE elements in miR-135a core promoter region, and verified estrogen-induced negative regulation of miR-135a expression. Previous studies of miR-135a were overexpressed in GCs from PCOS patients, and promoted GC proliferation and repressed GC apoptosis via repressing ESR2 expression [49]. Therefore, we provide a protection mechanism of estrogen-mediated TGFBR1 and CCND2 mRNA away from miR-135a mediated degradation. Meanwhile, these results may suggest that miR-135a has a different expression pattern during folliculogenesis compared with the pathological process. Under physiological or pathological accommodation, miR-135a and ESR2 may determine the state of GC growth via repressing each other’s expression (which merits further investigation).

## 5. Conclusions

In conclusion, our data document the roles of murine miR-135a in cell cycle retardation and cell proliferation suppression by suppressing *Tgfbr1* and *Ccnd2* expression. Our study suggests a pro-survival mechanism of ESR2/miR-135a/TGFBR1/CCND2 axis, which may provide a new method for improvement of female fertility.

## Figures and Tables

**Figure 1 cells-10-02104-f001:**
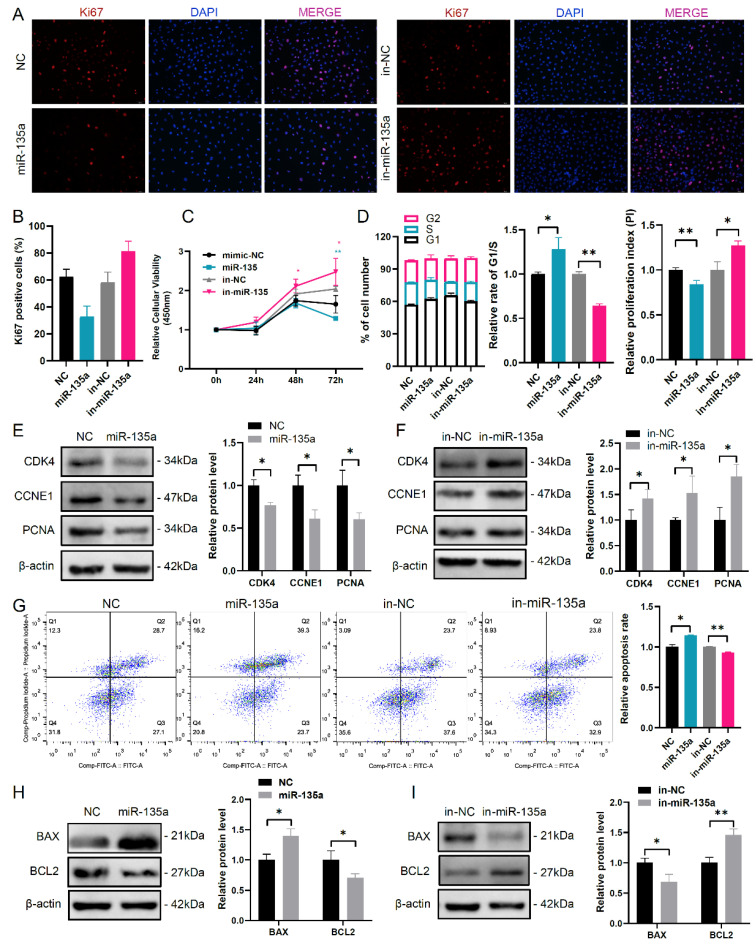
miR-135a represses cell proliferation in mGCs. (**A**) Immunofluorescence staining of cell proliferation marker Ki67 in mGCs. The nuclei were stained by DAPI (blue). Scale bar: 50 μm. miR-135a mimics is abbreviated to miR-135a. Negative control is abbreviated to NC. miR-135a inhibitor is abbreviated to in-miR-135a. Negative control inhibitor is abbreviated to in-NC. (**B**) The quantification of the Ki67-positive cells in (**A**). (**C**) Cell viability was detected by CCK-8 in mGCs. (**D**) The cell cycle of miR-135a was detected by flow cytometry assay in mGCs. The distribution of mGCs in each phase of cell cycle was shown in left panel. The relative rate of G1/S in mGCs was shown in middle panel. PI (proliferation index, PI = (G2 + S)/G1) was shown in right panel. Western blot analysis of (**E**) cell cycle and (**F**) proliferation-related protein levels in mGCs. (**G**) Assessment of apoptosis using Annexin V-FITC/PI and flow cytometry in miR-135 overexpressed and inhibited mGCs. Western blot analysis of apoptosis-related protein level in miR-135 (**H**) overexpressed and (**I**) inhibited mGCs. * *p* < 0.05 and ** *p* < 0.01 compared with controls according to two-tailed Student’s t test.

**Figure 2 cells-10-02104-f002:**
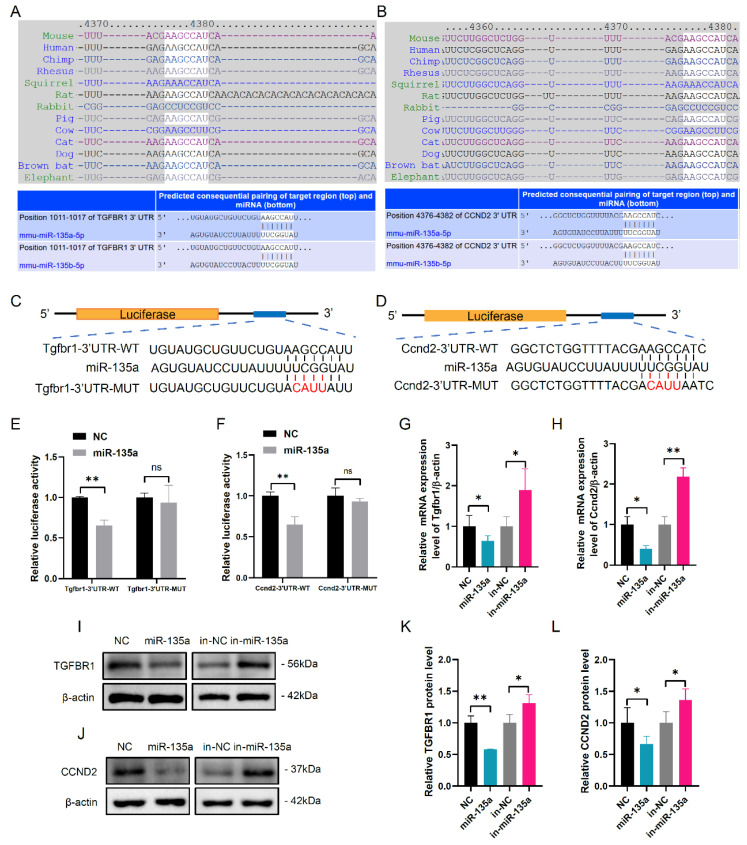
*Tgfbr1* and *Ccnd2* are the targets of miR-135a. Targetscan software showed the predicted miR-135a binding site in (**A**) *Tgfbr1* and (**B**) *Ccnd2* 3′UTRs. The wild type (WT) and mutational (MUT) 3′UTRs of (**C**) *Tgfbr1* and (**D**) *Ccnd2* mRNAs were shown in mGCs. The mutant bases in the 3’ UTR of target genes were shown with red font. The luciferase activity of recombinant (**E**) *Tgfbr1* and (**F**) *Ccnd2* 3′UTRs was determined in miR-135a overexpressed mGCs. NS, none significant difference. Relative mRNA level of (**G**) *Tgfbr1* and (**H**) *Ccnd2* was shown in miR-135 overexpressed and inhibited mGCs. Western blot analysis of (**I**) *Tgfbr1* and (**J**) *Ccnd2* was shown in miR-135a overexpressed and inhibited mGCs. The relative quantification of (**K**) TGFBR1 and (**L**) CCND2 protein levels was shown in the bar graph. Data were presented as mean ± SD (*n* = 3). * *p* < 0.05, ** *p* < 0.01.

**Figure 3 cells-10-02104-f003:**
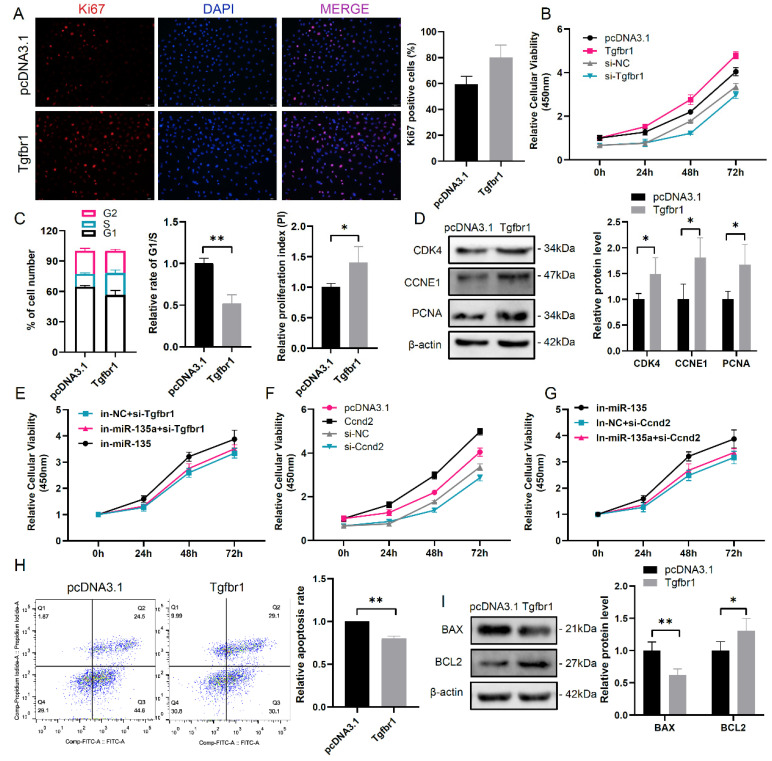
miR-135a affects mGC proliferation via regulating *Tgfbr1* expression. (**A**) Immunofluorescence staining of cell proliferation marker Ki67 in *Tgfbr1* overexpressed mGCs. The nuclei were stained by DAPI (blue). Scale bar: 50 μm. The quantification of the Ki67-positive cells in right panel. (**B**) Cell viability was detected by CCK-8 assay in *Tgfbr1* overexpressed or inhibited mGCs. (**C**) The cell cycle was detected by flow cytometry assay in *Tgfbr1* overexpressed mGCs. (**D**) Western blot analysis of cell cycle and proliferation-related protein levels in *Tgfbr1* overexpressed mGCs. (**E**–**G**) Cell viability was detected by CCK-8 assay with indicated treatment in mGCs. (**H**) The apoptosis assessment was measured using Annexin V-FITC/PI and flow cytometry in *Tgfbr1* overexpressed mGCs. (**I**) Western blot analysis of apoptosis-related protein level in *Tgfbr1* overexpressed mGCs. The means ± SD were calculated from three independent experiments. * *p* < 0.05, ** *p* < 0.01.

**Figure 4 cells-10-02104-f004:**
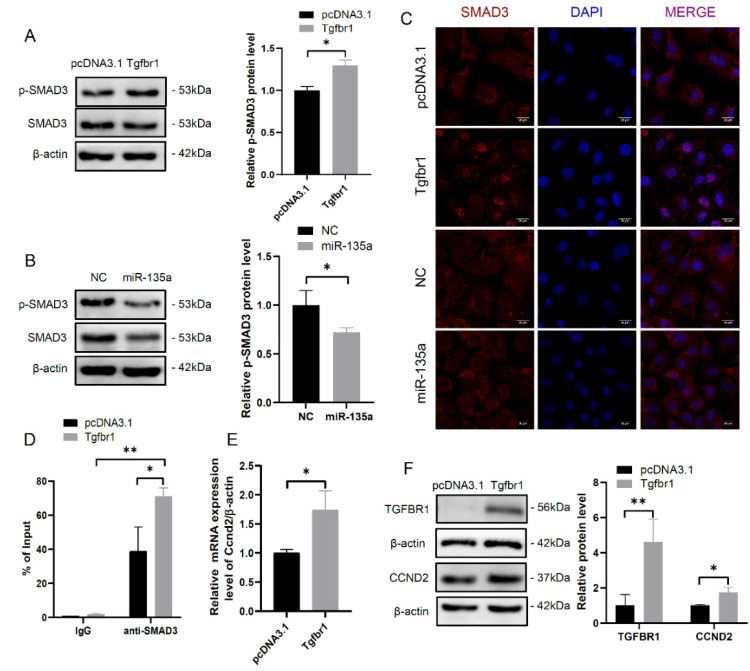
*Tgfbr1* enhances SMAD3-mediated *Ccnd2* promoter activity. (**A**) Western blot analysis of p-SMAD3 protein level in *Tgfbr1* overexpressed mGCs. The relative quantification of p-SMAD3 protein level in *Tgfbr1* overexpressed mGCs was shown in the right panel. (**B**) Western blot analysis of p-SMAD3 protein level in mGCs transfected with miR-135a mimics. The relative quantification of p-SMAD3 protein level in mGCs transfected with miR-135a mimics was shown in the right panel. (**C**) The subcellular distribution of SMAD3 was detected by immunofluorescence assay in mGCs treated with *Tgfbr1* overexpression or miR-135a mimics. Scale bar: 20 μm. (**D**) ChIP analyses of SMAD3 binding sites in *Ccnd2* promoter region followed by qPCR. (**E**) Relative mRNA level of *Ccnd2* in *Tgfbr1* overexpressed mGCs. (**F**) Western blot analysis of CCND2 protein level in *Tgfbr1* overexpressed mGCs. The means ± SD were calculated from three independent experiments. * *p* < 0.05, ** *p* < 0.01.

**Figure 5 cells-10-02104-f005:**
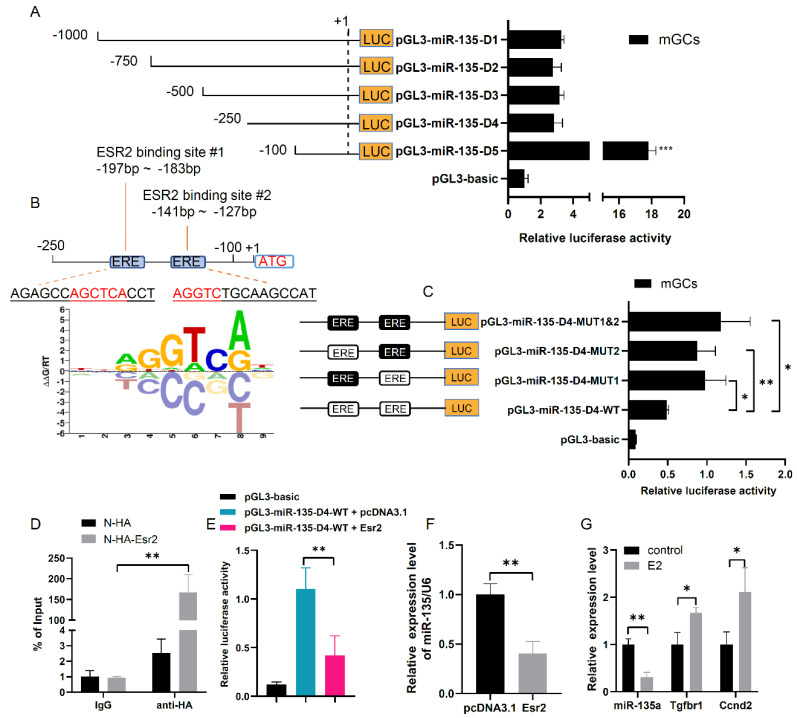
ESR2 negatively regulates miR-135a expression via binding to miR-135a promoter region. (**A**) Luciferase activity assay of a series of truncated miR-135a promoters in mGCs. (**B**) Schematic diagram of miR-135a promoter. (**C**) Site-directed mutation analysis of the miR-135a promoter using luciferase activity assays. Two intact ESR2 binding sites were indicated by empty boxes, respectively. The filled boxes showed the corresponding mutations. (**D**) ChIP analyses of ESR2 binding sites in miR-135a promoter region followed by qPCR. mGCs expressing N-HA were assayed as a control to exclude the possibility of nonspecific binding of anti-HA. (**E**) Luciferase activity of miR-135a promoter was downregulated in *Esr2* overexpressed mGCs. (**F**) Relative level of miR-135a in *Esr2* overexpressed mGCs. (**G**) Relative level of miR-135a, *Tgfbr1* and *Ccnd2* expression in estrogen-treated mGCs. The means ± SD were calculated from three independent experiments. * *p* < 0.05, ** *p* < 0.01, *** *p* < 0.001.

**Figure 6 cells-10-02104-f006:**
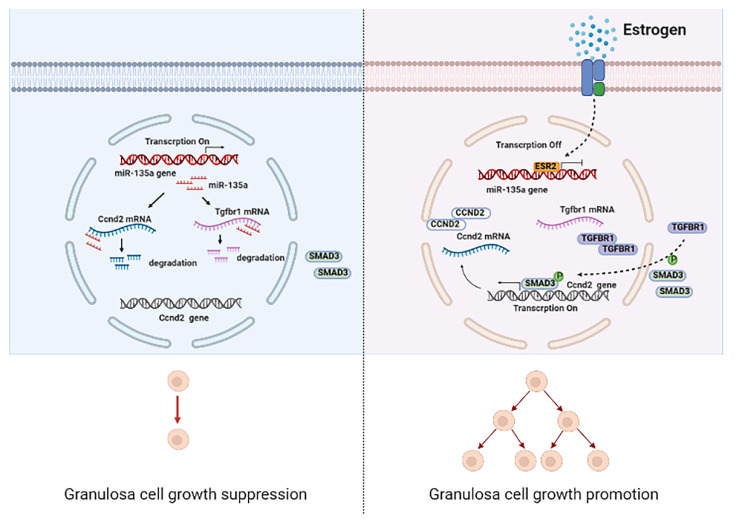
A schematic model for miR-135a regulating granulosa cell growth. The schematic diagram depicting the mechanism that miR-135a regulates murine granulosa cell growth. First, miR-135a binds to 3′UTR of *Tgfbr1* and *Ccnd2* to repress their expression. Then, downregulated TGFBR1 fails to facilitate nuclear distribution of SMAD3, which could not enhance *Ccnd2* expression. Finally, ESR2 functions as a transcription factor to directly bind to the miR-135a promoter region, decreases the transcriptional activity of miR-135a, and eventually suppresses miR-135a-mediated mGC growth suppression.

## Data Availability

The data that support the findings of this study are available from Fenge Li upon reasonable request.

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
