# Peer review of "miR-135a Suppresses Granulosa Cell Growth by Targeting *Tgfbr1* and *Ccnd2* during Folliculogenesis in Mice"

_cells, 2021, doi:10.3390/cells10082104_

Round 1

Reviewer 1 Report

In this study, the authors investigated the role of miR-135a in mGC growth. They reported that miR-135a inhibits cell cycle and proliferation in murine GCs via targeting Tgfbr1 and Ccnd2 genes in mGCs. They showed also that TGFBR1 regulates the transcription of Ccnd2 by facilitating the recruitment of SMAD3 to Ccnd2 promoter. Moreover, the authors proved that estrogen promotes the binding of ESR2 to the miR-135a promoter region and thus negatively regulates its endogenous expression to improve Tgfbr1 and Ccnd2 expression. Finally, they suggested a model for ESR2/miR-135a/TGFBR1/CCND2 network in regulating granulosa cell growth.

The study is well designed and presented clearly. It provides insights into understanding the regulatory mechanisms that regulate GC growth and proliferation.  

Some minor comments and corrections to be considered.

Introduction:

Line 69: Please indicate here the differences between the two breeds regarding the ovulation rate and litter size in correlation to miR-135a expression in the pre-ovulatory ovarian follicles. This was indicated in the results section (Line 161), however, it is better to explain it in the introduction as a base or a reason why you selected this particular miRNA.

MM:

Line 100: Some details about the transfection experiment are required or adding a reference for the transfection details.

Results:

In general, the quality of the figures needs to be improved. For example, the apoptosis assessment-related figures are blurred.  

Line 160: Mir-135a should be written with a capital R (MiR-135a). The same in Line 196

Line 166: Did the author count or quantify the signal of Ki67 and compare between the different groups? The presented results showed only representative photos.

Line 219: This sub-section needs some rearrangements to be consistent with the presented results in fig 3. For example, the results of Tgfbr1 and Ccnd2 are separated in the text however, both are on the same graph in the fig.

In the same section, did the authors analyse the immunofluorescence staining of cell proliferation marker Ki67 in Ccnd2 overexpressed mGCs as well?

Line 232-233: presenting the expression differences with fold change for elevation or reduction of the expression should be as 1.5 fold, for example, to indicate an increase or decrease by 50% from the control and as 2 fold in case of 100% change and so on. However, here the author mentioned it as 0.5 fold. Please check it and rewrite it either as a fold or as a percentage.

Line 282: do you mean here the wild-type pGL3-miR-135a-D5 or D4?

Discussion:

The authors identified that ESR2 negatively regulated miR-135a expression through binding to two ERE elements in the miR-135a core promoter region. However, another study reported the opposite (doi:10.1016/j.mce.2019.110478). They reported that miR-135a repressed ESR2 expression in GCs. It is necessary to discuss such results and to correlate them with the current findings.

Another study also reported the role of miR-135a in GCs (doi.org/10.1007/s43032-020-00155-0) which would enrich the discussion or the introduction as it is closely related to the current work.

Author Response

Thank you very much for the suggestions and comments on our manuscript. The point-by-point responses to the reviewer’s comments are provided. Please see the attachment.

Reviewer 2 Report

The manuscript describes a role for miR-135a in granulosa cell proliferation using a mouse model. The study is designed and carried out in a logical and clear manner.

The description of western blot in the materials and methods section is missing information related to the gel electrophoresis portion of the method.

In studies focused on tgfbr1, such as figure 3A, was any ligand present in the media? this should be clarified.

Figure 3 F and G:  this experiment needs to have miR-135 inhibition alone.  Although the effect shown in Fig 1 was referenced, the control needed to be included in those experiments to show the effect.

Author Response

(The authors gave the same response as above.)
